# Current Intraoperative Imaging Techniques to Improve Surgical Resection of Laryngeal Cancer: A Systematic Review

**DOI:** 10.3390/cancers13081895

**Published:** 2021-04-15

**Authors:** Lorraine J. Lauwerends, Hidde A. Galema, José A. U. Hardillo, Aniel Sewnaik, Dominiek Monserez, Pieter B. A. A. van Driel, Cornelis Verhoef, Robert J. Baatenburg de Jong, Denise E. Hilling, Stijn Keereweer

**Affiliations:** 1Department of Otorhinolaryngology, Head and Neck Surgery, Erasmus MC Cancer Institute, 3015 GD Rotterdam, The Netherlands; l.lauwerends@erasmusmc.nl (L.J.L.); h.galema@erasmusmc.nl (H.A.G.); j.hardillo@erasmusmc.nl (J.A.U.H.); a.sewnaik@erasmusmc.nl (A.S.); d.monserez@erasmusmc.nl (D.M.); r.j.baatenburgdejong@erasmusmc.nl (R.J.B.d.J.); 2Department of Surgical Oncology and Gastrointestinal Surgery, Erasmus MC Cancer Institute, 3015 GD Rotterdam, The Netherlands; c.verhoef@erasmusmc.nl; 3Department of Orthopedic Surgery, Isala Hospital, 8025 AB Zwolle, The Netherlands; p.b.a.a.van.driel@isala.nl

**Keywords:** surgical margins, laryngeal cancer, intraoperative imaging, narrow-band imaging, fluorescence imaging, Raman spectroscopy

## Abstract

**Simple Summary:**

Laryngeal cancer is a prevalent head and neck malignancy, with poor prognosis and low survival rates for patients with advanced disease. The recurrence rate for advanced laryngeal cancer is between 25 and 50%. In order to improve surgical resection of laryngeal cancer and reduce local recurrence rates, various intraoperative optical imaging techniques have been investigated. In this systematic review we identify these technologies, evaluating the current state and future directions of optical imaging for this indication. Evidently, the investigated imaging modalities are generally unsuitable for deep margin assessment, and, therefore, inadequate to guide resection in advanced laryngeal disease. We discuss two optical imaging techniques that can overcome these limitations and suggest how they can be used to achieve adequate margins in laryngeal cancer at all stages.

**Abstract:**

Laryngeal cancer is a prevalent head and neck malignancy, with poor prognosis and low survival rates for patients with advanced disease. Treatment consists of unimodal therapy through surgery or radiotherapy in early staged tumors, while advanced stage tumors are generally treated with multimodal chemoradiotherapy or (total) laryngectomy followed by radiotherapy. Still, the recurrence rate for advanced laryngeal cancer is between 25 and 50%. In order to improve surgical resection of laryngeal cancer and reduce local recurrence rates, various intraoperative optical imaging techniques have been investigated. In this systematic review, we identify these technologies, evaluating the current state and future directions of optical imaging for this indication. Narrow-band imaging (NBI) and autofluorescence (AF) are established tools for early detection of laryngeal cancer. Nonetheless, their intraoperative utility is limited by an intrinsic inability to image beyond the (sub-)mucosa. Likewise, contact endoscopy (CE) and optical coherence tomography (OCT) are technically cumbersome and only useful for mucosal margin assessment. Research on fluorescence imaging (FLI) for this application is sparse, dealing solely with nonspecific fluorescent agents. Evidently, the imaging modalities that have been investigated thus far are generally unsuitable for deep margin assessment. We discuss two optical imaging techniques that can overcome these limitations and suggest how they can be used to achieve adequate margins in laryngeal cancer at all stages.

## 1. Introduction

With almost 40,000 cases in Europe in 2018 [1], laryngeal cancer is the second most common malignancy of the head and neck region. Early diagnosis and adequate preoperative assessment increase the likelihood to cure disease while preserving functionality. Outcomes for patients with early stage (T1 and T2) tumors are favorable, with cure rates of 80–90% [2,3]. More advanced staged laryngeal squamous cell carcinoma (SCC) have been reported to have recurrence rates ranging between 25 and 50% [4]. These tumors are associated with loss of laryngeal function and poor prognosis, with 5-year survival rates dropping to 40% for patients with stage IV disease [2].

In head and neck cancer, margin status remains one of the most important factors for local recurrence and overall survival, demonstrating the need to improve surgical resection. In patients undergoing (total) laryng(-opharyng)ectomy, inadequate margins are clearly associated with poor survival [5,6]. In patients that are treated with laser surgery (generally early stage laryngeal cancer), the relationship between resection margin and patient outcome is less evident. Numerous studies on laryngeal cancer produce inconsistent results with regards to margin status and local control or relapse rate [7,8,9,10]. This is caused by both post-resection and -fixation shrinkage of the tissue specimen and cauterization artefacts associated with laser carbonization, which hamper an adequate histological evaluation of the surgical margins in those cases [11]. Consequently, a positive margin on the tissue specimen may not always indicate that there are residual tumor cells in the wound bed [12].

Importantly, it was clearly demonstrated that tumor-positive additional wound bed biopsies, taken after transoral CO2 laser microsurgery (TLM) of laryngeal cancer, significantly impacted local control [12]. This finding illustrates that there is a clear clinical need for improved intraoperative margin assessment during laser surgery. Extensive wound bed sampling is no good alternative because this only allows analysis of a fraction of the wound bed, making it prone to sampling error.

In addition, margin adequacy cannot be determined from sampling the wound bed. When a total laryng(-opharyng)ectomy is performed in advanced stage laryngeal cancer, a specimen-driven analysis of the surgical margins will likely be better, as has been shown in similar cases of oral cancer surgery [13,14]. Ideally, margin status would be assessed during surgery through less invasive and more thorough inspection measures. Improved intraoperative margin assessment may also facilitate more conservative resection during partial laryngectomies and prevent superfluous biopsies, which are important considerations in improving final functional outcome and organ preservation [11].

The surgeon’s ability to distinguish between malignant and healthy tissue during laryngeal surgery remains suboptimal, elucidating the need for objective intraoperative tools for margin assessment that facilitate both specimen and wound bed evaluation. Optical imaging has long been investigated and implemented for the early diagnosis of laryngeal cancer. Over the years, a number of these imaging modalities have been suggested as intraoperative tools to facilitate real-time enhanced visualization of resection margins [15,16]. This systematic review provides an overview of the intraoperative imaging modalities that have been studied to improve surgical resection of laryngeal tumors.

## 2. Materials and Methods

This systematic review is reported according to the Preferred Reporting Items for Systematic Reviews and Meta-Analyses (see PRISMA 2020 checklist, Appendix A), and was registered in the PROSPERO database for systematic reviews (CRD42020187479).

### 2.1. Literature Search and Study Selection

A systematic search was conducted in the Embase, Medline, Web of Science Core Collection, Cochrane, and Google Scholar databases. A search strategy was constructed using the terms ‘laryngeal cancer’ and ‘intraoperative imaging’. Appendix A shows the full search strings per database. The last search was conducted on 26 October 2020. Based on the title and abstract, all clinical studies concerning intraoperative imaging techniques in laryngeal cancer surgery and diagnosis were included. Only journal articles in English with the full text available were considered. Next, articles were screened on full text for final inclusion. Articles that did not explicitly describe imaging techniques for margin assessment during laryngeal cancer surgery were excluded. Screening on title and abstract and on full text was done by two independent authors (L.J.L. and H.A.G.). Disagreements were discussed with a third author (S.K.).

### 2.2. Data Extraction

The following data were extracted from Tables and text: imaging technique, application of imaging technique (image-guided resection and wound bed assessment), T-stage, surgical procedure, negative margin definition, negative margin rates, extra resection rate, patient-related outcomes (recurrence rate, disease specific survival, recurrence free survival), and diagnostic outcomes (sensitivity, specificity, positive predictive value, and negative predictive value).

### 2.3. Quality Assessment

The ‘methodological index for non-randomized studies’ (MINORS) was applied to all studies in the final inclusion [17]. The MINORS is a validated methodological index for non-randomized studies in which the included studies are rated on 8 (non-comparative studies) or 12 items (comparative studies). All items can obtain a score between 0 and 2 points. A 0 points score indicates that it was not reported in the article, 1 point if it was inadequately reported, and 2 points if it was adequately reported. Hence, a maximum score of 16 (non-comparative studies) or 24 points (comparative studies) could be obtained.

## 3. Results

Through the search strategies, 4128 articles were identified. A total of seven additional articles were found through snowballing. After removing duplicates, 2644 articles were screened on title and abstract. Then, 68 articles were screened on full text and 19 articles were included for qualitative assessment in the systematic review. After careful consideration, the contributions from articles with a MINORS score of 4 and lower were deemed of insufficient quality, and were excluded from the systematic review. This additional screening resulted in 18 studies for final inclusion. Appendix A shows the PRISMA flow diagram. All studies were non-randomized studies. Study characteristics and main outcomes are shown in Table 1. These studies reported on the following imaging techniques: narrow-band imaging (10), autofluorescence (3), fluorescence (2), contact endoscopy (2), and optical coherence tomography (1). MINORS-index from the included articles ranged from 5 to 19. The full MINORS of the included articles is shown in Appendix A.

### 3.1. Intraoperative Narrow-Band Imaging

Narrow-band imaging (NBI) is an endoscopic technique that uses selective light wavelengths to visualize the abnormal vascular patterns of the mucosa associated with premalignant or neoplastic lesions. NBI uses optical filters to select blue and green light (wavelengths of 415 and 540 nm, respectively). Narrow-band blue light, because of its low penetration depth, highlights mainly the mucosal surface, making subjacent capillaries appear brown. At the same time, the narrow-band green light with its higher penetration depth, allows it to enhance visibility of vessels in the submucosa, making them appear dark green (Figure 1A and Figure 2A,B).

The value of NBI during the diagnostic phase for early detection of laryngeal cancer and its precursor lesions is undisputed [36]. Various studies have demonstrated that a combination of NBI and white light (WL) endoscopy is superior to WL alone for its diagnostic capability [36,37,38,39,40]. Ni et al., for example, reported a diagnostic accuracy of 91% for NBI, compared with 71% for standard endoscopy [39]. Detection sensitivity of laryngeal cancer and its precursor lesions in particular is increased in multiple studies, with values ranging between 95 and 100%, confirming its ability to easily distinguish (pre)malignant from benign lesions [22,36,37,38,39,40,41]. NBI has been reported to identify malignant lesions that were missed by WL alone [26,40], and, conversely, has also been reported to reduce the amount of suspicious lesions compared to WL [37]. NBI is redundant for diagnosing advanced tumors, as those can be adequately visualized with conventional WL endoscopy. Nevertheless, NBI can still define tumor borders with a diagnostic gain of 11% [38], which is an important quality when considering NBI for intraoperative margin assessment.

Beyond the diagnostic phase, NBI holds a lot of promise for achieving more accurate margin delineation during TLM for early glottic cancer (Table 2 and Table 3) [42]. It has been reported that NBI was able to, statistically, significantly reduce [3,18,19,23,25], or completely prevent, the occurrence of positive superficial margins [20,21]. Overall, two studies reported the detection of additional lesions that were not seen with WL alone [20,21]. Additionally, four studies show significantly lower recurrence rates and better recurrence-free survival (RFS) with NBI as compared to WL alone [3,20,21,22]. The majority of studies conclude superiority of NBI over WL endoscopy in patients with early laryngeal cancer, with Klimza et al. suggesting that the added value of NBI is higher in T2 tumors, as compared to T1 tumors [20]. Then, one study included 3 patients with advanced (T3–T4) cancer, without differentiating the results according to tumor stage [25]. Despite the anatomical restraints present in the resection of advanced laryngeal tumors, Šifrer et al. reported a significantly higher rate of resection margins that were initially tumor negative in patients treated with NBI compared to their historical control group, comprising all patients previously operated by the same surgeon, before availability of NBI instrumentation [23]. The proportion of patients with T3 or T4 stage tumors, however, was not specified. Piersala et al. evaluated intraoperative NBI in patients with T2–T3 cancer, successfully achieving more accurate superficial resection margins [21]. Finally, Zwakenberg et al. recently reported that they were able to identify tumor extent by an increase of 5.7% with NBI compared to WL (*p* = 0.02) [26]. Nonetheless, these findings were not biopsy confirmed. Additionally, while this study reported inclusion of patients with Tcis-T4 tumors, the proportions were left unspecified and outcomes were not correlated with the tumor stage.

Overall, NBI was found to modify intraoperative decision-making in several ways: better visualization of tumor borders led to more secure resection margins, detection of additional lesions prompted additional biopsies, and NBI-led upstaging resulted in changes to the surgical plan [24]. With its established position in current clinical practice for early diagnosis of laryngeal cancer, NBI is the most prominently featured modality in research of optical modalities for intraoperative margin detection. An important benefit of NBI is that it is not a labeling technique requiring exogenous agents, therefore side effects or complications are of no concern, nor does it require laborious agent development. Nevertheless, NBI is specifically suited to identification of superficial carcinomas due to their aberrant vascular pattern, thus limited to detection of mucosal margins [8,43].

### 3.2. Intraoperative Autofluorescence Imaging

Autofluorescence (AF) is the natural fluorescence emission of tissue when exposed to light of a suitable wavelength (Table 3). AF is based on the common presence of intrinsic biomolecules acting as endogenous fluorophores [44]. Upon illumination, the electrons of a fluorophore are elevated to a higher energy level. The fluorophore is unstable in this excited state, and will rapidly revert to a slightly lower, more stable energy level by expelling heat. With some of the energy already released, the emitted light has a longer wavelength and is of lower energy than that of the illuminating light [15].

The fundamental principle of AF is related to changes in tissue morphology, optical properties, and the concentration of endogenous fluorophores in tissue as a result of neoplasia. When exposed to blue light, healthy laryngeal mucosa emits green fluorescence, whereas neoplastic mucosa fluoresces red-violet (Figure 1B and Figure 2C,D). The advantage of AF over conventional WL endoscopy stems from the fact that (pre)malignant lesions can be differentiated from normal tissue because of decreased AF [15,16,45]. AF has previously been widely evaluated for the early diagnosis of laryngeal cancer [46], demonstrating demarcation of tumor borders and high sensitivity compared to standard technique (97.5% vs. 82.5%) [47].

In total, three clinical studies evaluate the intraoperative benefit of AF for margin assessment. Succo et al. reported high sensitivity and specificity (96.8% and 98.5%), stating an improvement in diagnostic accuracy over WL alone in 12 out of 73 cases (16.4%) [29]. Paczona et al. performed a more extensive excision based on AF in 2 of 10 cases, and reported a sensitivity and specificity of 97.4% and 60%, respectively [28]. In 2006, Fielding et al. reported the finding of additional malignant lesions in 5 out of 48 patients with laryngeal cancer, noting that AF may have potential for detecting unknown primaries and limiting the amount of sites requiring biopsies [27].

These studies suggest that direct AF can have a positive impact on disease-free survival and local control. However, more substantive studies are warranted to examine the added benefit of AF for more discrete identification of tumor presence, or positive margin reduction. AF is limited to examination of surface tissues. The illuminating light does not penetrate diseased epithelium, so that neoplastic changes within the basal mucosal layer may be hidden by epithelial hyperkeratosis. The presence of necrosis, scar tissue, bleeding, and inflammation can alter mucosal fluorescence in an unpredictable manner [15].

### 3.3. Intraoperative Fluorescence Imaging

Fluorescence Imaging (FLI) is an emerging optical modality that facilitates intraoperative guidance aimed at complete resection of tumor tissue [48]. After systemic administration of a fluorescent agent and an appropriate dispersion time interval, FLI is performed real-time using an intraoperative camera system (Figure 1C and Figure 2C,E).

The application of FLI for intraoperative margin detection in laryngeal cancers is thus far underexplored. A 2004 trial reported a sensitivity of 96% for 5-aminolevulinic acid-induced protoporphyrin-IX (5-ALA Pp-IX) fluorescence in the control of pharyngo-laryngeal cancer [30]. In this study including 13 patients with T1–T2 laryngeal cancer, Pp-IX fluorescence better visualized the borderline of superficial neoplastic tissue, facilitating more precise diagnosis and excision.

The attenuating effects of light scattering, non-specific autofluorescence and absorption are relatively low for fluorescence in the near-infrared (NIR) range [49]. In a study on the use of intravenous administration of NIR agent indocyanine green (ICG) for therapeutic lymph node dissection, Digonnet et al. briefly described primary tumor examination 3 patients with laryngeal cancer, reporting a fluorescent signal in 2 patients [31]. FLI with tumor-specific NIR fluorescent agents has not yet been investigated in laryngeal cancer (Table 3).

### 3.4. Contact Endoscopy

Contact endoscopy (CE) is a technique that allows for in vivo visualization in cellular detail. After staining superficial cells of the mucosa with methylene blue, a microscopic endoscope is placed in direct contact with the surface, providing images of the suspected area at ×60 or ×150 magnification (Figure 3A,B, Table 3). Contact laryngomicroscopy is a non-invasive method capable of obtaining detailed information of cells and blood vessels in the live epithelium without biopsy, distinguishing malignant from benign tissue. CE has been shown to detect post-radiotherapy residual and recurrent cancer in the upper aerodigestive tract with similar sensitivity and specificity compared to in non-irradiated tissues [50].

CE has been studied for the detection of microscopic margins during surgery in 10 patients with early glottic SCC, yielding histopathologically confirmed complete excision (negative margins defined as ≥2 mm) by Dedivitis et al., without local recurrence after a minimum follow-up period of 5 years [32]. This survival benefit in early laryngeal cancer patients is promising, but needs confirmation from larger scale, adequately powered studies. A more recent study in 43 patients with T1–T2 vocal fold carcinoma finds a similar, if slightly lower, negative resection rate for CE compared to frozen sections after histopathological examination of paraffin sections: 92% vs. 98.3% [33]. While these results are indicative of its reliability for early laryngeal cancer, studies including patients with advanced laryngeal lesions are absent. Additionally, CE requires topical staining and is associated with considerable instrumentation costs, requires a steep learning curve for the interpretation of the vascular network, and is solely capable of characterizing the superficial layers of the mucosa [51].

### 3.5. Optical Coherence Tomography

Optical coherence tomography (OCT) is the optical counterpart of ultrasound, and is capable of imaging cross-sectional anatomy at high resolutions in living tissue (Figure 3C,D, Table 3). OCT is based on changes in refractive index, by detecting light that is backscattered off tissue boundaries [15]. It is a noninvasive imaging method that allows in vivo use, provides real time information, and does not cause any side effects.

OCT has only been studied for intraoperative margin assessment during TLM by Shakhov et al. in 26 patients with early laryngeal carcinoma [34]. While they reported tumor margins extending beyond those seen with WL alone, no data were provided on diagnostic accuracy.

## 4. Discussion

There is a paucity of trials that analyze intraoperative imaging as a surgical adjunct during procedures for laryngeal cancer. The diagnostic value of NBI and AF in early laryngeal cancer is known. Their intraoperative application as an aid in defining surgical margins has not yet been studied as extensively, with most data available on NBI in this setting. As NBI visualization is limited to microvascular changes of the (sub)mucosa, evaluation of deep margin infiltration is impossible. Nevertheless, some recent studies report encouraging results of intraoperative NBI in patients with moderate-advanced laryngeal tumors. Distinct from intraoperative margin assessment, an interesting application for NBI is the follow-up of patients with early to moderate stage glottic cancer after curative treatment. Witkiewicz et al. recently reported that NBI after TLM is a good predictor of margin positivity, supporting clinical decision-making regarding the need for second-look surgery [53]. NBI during outpatient transnasal endoscopy was found to be an excellent follow-up tool of laryngeal cancer patients after radiotherapy to screen for recurrent disease [54]. Still, caution is warranted in the interpretation of NBI images. Post-radiotherapy nonspecific inflammation often presents as brownish spots similar to those that are typical for early cancer, and may thus be mistaken for recurrence [40,55]. More clinical studies are required to make conclusive statements about NBI for TLM of moderate-advanced laryngeal tumors. AF has been demonstrated to be useful in early laryngeal cancer detection, although intrinsic mucosal fluorescence can unpredictably be altered by numerous tissue properties [15].

Both OCT and CE are associated with various technical impracticalities. CE requires topical staining and is only able to clearly image the superficial layers of the epithelium [45]. Previous ex vivo and probe-based in vivo diagnostic efforts using OCT for laryngeal cancer have demonstrated the technique’s performance in distinguishing normal from cancerous tissue [52,56,57,58,59]. Nevertheless, its maximum penetration depth of ~2 mm restricts its use to peripheral margin assessment. This inherent limited applicability likely explains the apparent shortage of studies using AF, OCT, and CE for intraoperative margin assessment in the larynx. This systematic evaluation of imaging techniques for intraoperative margin assessment reveals a prevailing incompatibility of the studied modalities in their utility for advanced stage laryngeal cancer, primarily attributable to imaging depth limitations. Furthermore, our assessment based on the MINORS criteria has revealed that the methodology and scientific evidence of a sizable portion of included studies are of suboptimal quality (Appendix A). In view of this apparent research gap, it may be useful to investigate other, more widely applicable imaging techniques to improve surgical resection of both early and advanced laryngeal cancer.

Although only 2 studies employing FLI for margin assessment were identified, laryngeal cancer surgery might be amenable to the more prevailing developments in FLI. An important advantage of FLI is that it can be used to scan large surfaces. In addition to the possibility to image fluorescence at considerable depths and tissues other than the mucosa, FLI is thus ideally suited for wound bed inspection and detection of clinically occult lesions. An interesting area to explore for FLI would be to identify suitable tumor-specific tracers for intraoperative imaging of early to advanced laryngeal cancer [60]. Previous research on the diagnostic value of FLI for laryngeal cancer is scarce, and, thus far, only features 5-ALA and ICG [46,61,62]. While these agents are easily available and approved by the U.S. Food and Drug Administration (FDA), they are both nonspecific. Furthermore, 5-ALA-induced Pp-IX fluorescence is outside the NIR-range, such that the signal is prone to significant interference due to the tissue’s optical characteristics [30]. The compatibility of FLI with CO2 laser surgery in head and neck cancer was recently validated by Odenthal and colleagues [63]. In vitro and preclinical results from this study suggest the effectiveness of fluorescence-guided surgery (FGS) using NIR dyes. Moreover, they demonstrated that laser cauterization and its associated autofluorescence are unlikely to interfere with the fluorescence signal of the injected tumor-specific tracer. Additionally, FLI for intraoperative margin assessment in head and neck cancers is currently a prolific field of research, with numerous phase I/II trials on oral cancer completed and ongoing (Figure 4) [64,65,66,67]. Oral cancer is well-suited to FGS, as the oral cavity is easily accessible to the NIR camera. However, with the rapidly evolving FLI cameras it is only a matter of time before FGS can be adopted to treat a wider variety of cancers. This notwithstanding, tumor-specific FLI is still under development and not without pitfalls. Signal processing for FLI is complex, and objective evaluation and quantification of both fluorescence and its influencing factors is challenging [49]. Development of suitable fluorescent agents is costly and time-consuming, with a lot of work left before regulatory approval and clinical implementation can be realized. Altogether, yet undiscovered potential for FLI in laryngeal cancer surgery at all stages is strongly hinted at, encouraging dedicated research for this indication.

As per the confines of this systematic review, there currently is no real-time application for Raman spectroscopy (RS) in margin assessment of laryngeal cancer. However, its demonstrated potential for both in and ex vivo tissue discrimination is worth noting. RS is a technique based on spectral differences between normal and malignant tissue, and is capable of near-instant, accurate, and non-invasive analysis of a tissue’s molecular composition (Table 3). In oral cancer specimen, water concentration analysis with RS has been demonstrated to objectively localize tumor borders [68]. The capacity of RS to differentiate tumor from healthy cells has been thoroughly demonstrated using laryngeal tissue samples, producing significant differences in Raman spectra between malignant and normal tissue [69,70,71,72]. Bergholt et al. were the first to demonstrate the feasibility of RS in transnasal endoscopic applications [73]. In their 2012 study, they acquired near-instant information of the endogenous biomolecular tissue composition from the larynx in healthy subjects, using a fiber-optic Raman probe. In 2016, Lin et al. conducted a similar study for in vivo probe-based RS during endoscopic examination, reporting improved diagnosis [74]. In a recent study of Stimulated Raman Scattering (SRS), Zhang et al. demonstrate the efficacy of deep-learning based SRS microscopy for margin assessment through simulated resection margins on completely removed larynx specimen [75]. Contrary to NBI and AF, RS is able to characterize tissue other than the mucosa. This is an important advantage allowing for post-resection assessment of tumor extension in the deep plane [8]. Nonetheless, RS is limited to point measurements, requiring direct tissue contact. As wide-field imaging is impossible, RS cannot be employed to screen for occult lesions and second primary tumors, and resection guidance and complete wound bed inspection are not currently feasible. The strength of RS lies in its application for specimen-driven margin assessment, at uniquely high specificity.

## 5. Conclusions

The demonstrated effectiveness of NBI in the early diagnosis of laryngeal cancer has led to reasonable evidence for its application in intraoperative margin detection, although inclusion of patients with T3–T4 laryngeal cancer is rare. An important disadvantage remains the technique’s intrinsic inability to image structures other than the (sub-)mucosa. Similar limitations hold for AF, OCT, and CE. Finally, FLI has scarcely been studied for intraoperative margin assessment in laryngeal cancer. Indeed, one might need to consider optical imaging advancements in adjacent anatomical regions and similar applications to gain insight into which modalities warrant further exploration in achieving adequate margins. NIR FLI with tumor-specific fluorescent agents and RS are emerging technologies that overcome many of the limitations associated with NBI, AF, CE, and OCT. With evidence supporting their value in assessing tumor depth infiltration, we believe it is worthwhile to pursue their clinical development for margin assessment in laryngeal cancer.

## Figures and Tables

**Figure 1 cancers-13-01895-f001:**
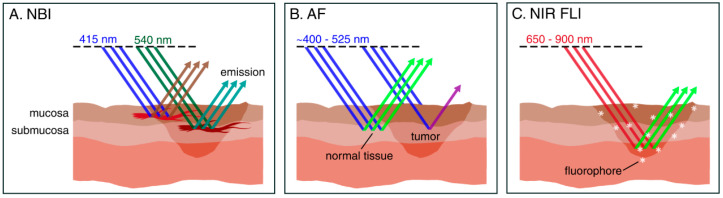
Schematic representation of fluorescence-based imaging working principles. (**A**) Narrow-band imaging (NBI): NBI light consists of two wavelengths: 415 nm blue and 540 nm green light. The blue light penetrates the mucosa, increasing contrast in these capillaries (appearing brown), whereas the green light reaches the submucosa, visualizing the vasculature in these deeper layers (appearing dark green). (**B**) Autofluorescence (AF) imaging: When exposed to blue light, healthy laryngeal mucosa emits green fluorescence, whereas neoplastic mucosa emits both light of lower energy (red-violet) and heat. (**C**) Near-infrared (NIR) fluorescence imaging (FLI): When exposed to NIR light, previously administered fluorophores emit light of a wavelength, which is specific to the fluorophore (e.g., excitation and emission wavelengths for ICG: ~780 and 820 nm, respectively). Penetration depth of light in the NIR spectrum ranges from 0.5 to more than 10 mm, contingent to the wavelength.

**Figure 2 cancers-13-01895-f002:**
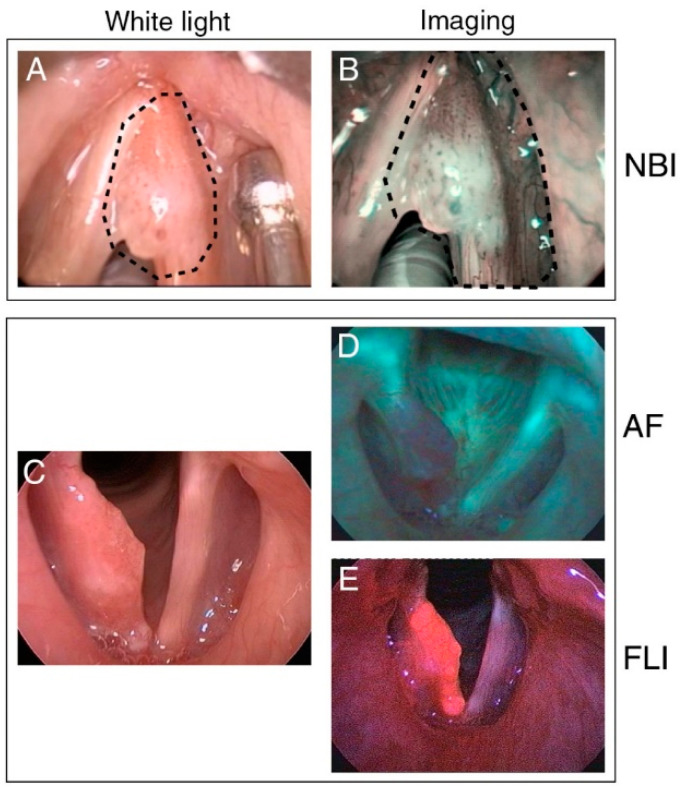
Clinical examples of fluorescence-based imaging modalities in laryngeal cancer. (**A**) White light image of cT1a laryngeal squamous cell carcinoma of the anterior third of the right vocal fold and its presumptive resection margins (dotted line) compared to (**B**) Narrow-Band Imaging (NBI), showing thick dark spots typically associated with carcinoma in situ. The planned superficial resection margins of type II cordectomy (**A**) were extended on the basis thereof (**B**). (**C**) White-light laryngoscopy demonstrating a bulging small squamous cell carcinoma of the right vocal fold with extension from the anterior commissure to the vocal process. (**D**) Autofluorescence (AF) laryngoscopy shows a reddish-violet fluorescence of the cancerous right vocal fold and green fluorescence of the normal mucosa. (**E**) Protoporphyrin IX accumulates in the cancerous tissue demonstrating a bright orange-red fluorescence signal during indirect laryngoscopy. (**A**,**B**) are adapted and reproduced from Garofolo et al. [18], by permission of SAGE Publications, Inc. (**C**–**E**) are adapted and reproduced from Arens et al. [35], by permission of Springer Nature.

**Figure 3 cancers-13-01895-f003:**
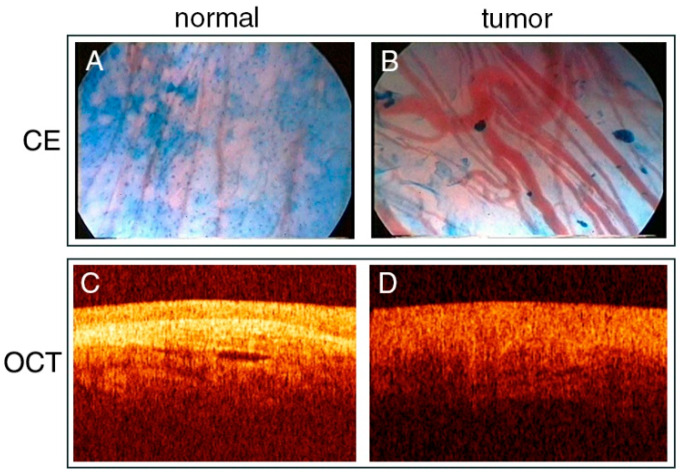
Clinical examples of contact endoscopy (CE) and optical coherence tomography (OCT) in laryngeal cancer. (**A**) Normal vascular network of the vocal fold in CE. (**B**) Vascular network of the vocal fold carcinoma in CE. (**C**) OCT image of healthy portion of larynx. (**D**) OCT image of exophytic tumor of larynx. A and B were adapted and reproduced from Stefanescu et al. (CC BY 4.0) [33]. C and D were adapted and reproduced from Sergeev et al. (CC BY 4.0) [52].

**Figure 4 cancers-13-01895-f004:**
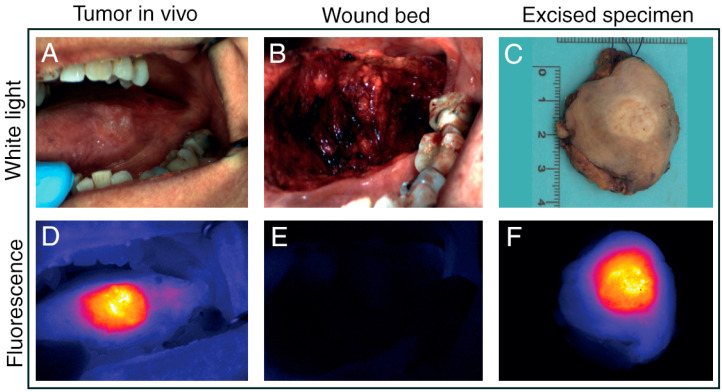
Fluorescence imaging using pH-activatable nanoprobe ONM-100 in head and neck squamous cell carcinoma of the tongue. In vivo (**A**,**B**,**D**,**E**) and ex vivo (**C**,**F**) visualization of high fluorescence signal in the tumor and low fluorescence signal in tumor-free margin and wound bed. Adapted and reproduced from Voskuil et al. (CC BY 4.0) [67].

**Table 1 cancers-13-01895-t001:** Included articles.

First Author	Year	Imaging Technique ^4^	Control Group	Study Design	T-Stage	*n* = (Imaging)	*n* = (Control)	Surgical Procedure	Application of Imaging	Negative Margin Definition
Fiz [3]	2017	NBI	WL ^2^	Retrospective	Tis-T2	311	323	TLM	Pre-resection	≥1 mm
Garofolo [18]	2014	NBI	WL ^2^	Prospective	Tis-T1a	82	152	TLM	Pre-resection	≥1 mm
Hainarosie [19]	2019	NBI	WL ^1^	nr	nr	23	23 ^1^	TLM	Post-resection	nr
Klimza [20]	2019	NBI	na	Retrospective	T1-T2	44	na	TLM	Pre-resection	≥5 mm
Piersiala [21]	2018	NBI	na	Prospective	T2-T3	98	na	TLM	Pre- and post-resection	≥3 mm
Plaat [22]	2017	NBI	WL ^2^	Retrospective	Tis-T2	42	51	TLM	Pre- and post-resection	nr
Šifrer [23]	2017	NBI	WL ^2^	Prospective	nr ^3^	14	8	Laryngectomy	Pre- and post-resection	nr
Srivastava [24]	2016	NBI	na	Retrospective	Tis-T2	30	na	TLM	Pre-resection	nr
Vicini [25]	2015	NBI	WL	Prospective	T1-T4	7	4	Transoral surgery	Pre- and post-resection	≥2 mm
Zwakenberg [26]	2021	NBI	WL ^1^	Prospective	Tis-T4	89	89 ^1^	TLM, TLE	Pre-resection	nr
Fielding [27]	2006	AF	na	Prospective	nr	48	na	TLM	Pre-resection	nr
Paczona [28]	2003	AF	WL ^1^	Prospective	nr	10	10 ^1^	TLM	Pre- and post-resection	nr
Succo [29]	2014	AF	WL ^1^	Prospective	T1-T2	73	73 ^1^	TLM	Pre-resection	≥1 mm
Csanády [30]	2004	FLI (5-ALA)	na	nr	T1-T2	13	na	TLM	Pre- and post-resection	nr
Digonnet [31]	2016	FLI (ICG)	na	Prospective	T1-T4	3	na	Laryngectomy	Pre- and post-resection	nr
Dedivitis [32]	2009	CE	na	Prospective	T1b-T2	10	na	Open surgery	Pre-resection	≥2 mm
Stefanescu [33]	2016	CE	na	nr	T1-T2	43	na	TLM	Pre-resection	nr
Shakhov [34]	2001	OCT	na	nr	Tis-T2	26	na	TLM	Pre-resection	nr

NBI = narrow-band imaging; AF = autofluorescence; FLI = fluorescence imaging; 5-ALA = 5-aminolevulinic acid; ICG = indocyanine green; CE = contact endoscopy; OCT = optical coherence tomography; WL = white light; TLE = total laryngectomy; TLM = transoral laser microsurgery; na = not applicable; nr = not reported. ^1^ The patients served as their own controls: first the regular treatment was performed, followed by extra imaging. Based on extra imaging, extra resection was done. ^2^ Historical control group. ^3^ T-stage was not specified, but article text indicates that only patients with advanced cancer treated by laryngectomy were included. ^4^ For NBI, the separate outcomes are further described and compared in Table 2.

**Table 2 cancers-13-01895-t002:** Narrow-band imaging (NBI) trial data.

Article	T-Stage	*n* = NBI ^1^/WL	Negative Margin Rate ^2^ NBI (vs. WL)	Patient-Related Outcome NBI (vs. WL)	Diagnostic Outcome NBI (vs. WL)	Other Findings Related to NBI
Fiz [3]	Tis-T2	311/323	50% (30%)	*RFS*^3^: 83.9% (78.9%) *DSS*^4^*:* 98.7% (98.8%)	nr	nr
Garofolo [18]	Tis-T1a	82/152	96.4% (76.3%)	nr	nr	nr
Hainarosie [19]	nr	23/23	98% (58.8%)	nr	nr	nr
Klimza [20]	T1-T2	44/na	100%	*Local recurrence:* 3/44: 6.8%	nr	Additional lesions not seen with WL
Piersiala [21]	T2-T3	98/na	100%	*Local recurrence:* 5/98: 5.10%	*NBI + WL:* Sens ^5^: 100% Spec ^6^: 98.88% PPV ^7^: 90% NPV ^8^: 100% Accuracy: 98.98%	Additional lesions not seen with WL in 10.2% of patients
Plaat [22]	Tis-T2	42/51	nr	*Local recurrence:* 2% (24%) *Two-year RFS:* 98% (82%)	nr	nr
Šifrer [23]	T3-T4	14/8	88.9% (70.9%)	nr	*NPV*: 95.9% (88.4%)	nr
Srivastava [24]	Tis-T2	30/na	nr	nr	nr	Upstaging TNM class
Vicini [25]	T1-T4	7/4	87.9% (57.9%)	nr	Sens: 72.5% Spec: 66.7% NPV: 87.9%	nr
Zwakenberg [26]	Tis-T4	89/89	nr	nr	Sens: 95% Spec: 82% PPV: 87% NPV: 92% Accuracy: 89%	5.7% increase in identified tumor extent with NBI compared to WL (*p* = 0.02)

nr = not reported; na = not applicable; WL = white light. ^1^ NBI: narrow-band imaging. ^2^ Percentage of negative resection margins obtained with NBI (vs. with WL). ^3^ RFS: recurrence-free survival. ^4^ DSS: disease-specific survival. ^5^ Sens: sensitivity. ^6^ Spec: specificity. ^7^ PPV: positive predictive value. ^8^ NPV: negative predictive value.

**Table 3 cancers-13-01895-t003:** Working principles and clinical usability of all imaging techniques for intraoperative margin assessment in laryngeal cancer.

Imaging Technique	Working Principle	Clinical Usability for Intraoperative Margin Assessment: Pros (+) and Cons (−)
**Narrow-Band** **Imaging** **(NBI)**	NBI uses blue (415 nm) and green (540 nm) light corresponding with the main peak absorbance of hemoglobin, to enhance visibility of mucosal and submucosal capillaries, respectively.	+ widely studied + provides real-time information + does not require exogenous agents + can be used to scan large surfaces for occult tumors + suitable for mucosal margin delineation (pre-resection) − not suitable for deep margin assessment
**Autofluorescence (AF)**	AF detects changes in tissue morphology and optical properties as a result of neoplasia. Using blue light, AF can differentiate between healthy and neoplastic laryngeal mucosa.	+ provides real-time information + does not require exogenous agents + can be used to scan large surfaces for occult tumors − not suitable for deep margin assessment
**Fluorescence** **Imaging** **(FLI)**	FLI uses a systemically administered fluorescent agent that ‘targets’ tumor cells over healthy cells. A dedicated camera system is required to detect these fluorescent agents, facilitating real-time image-guided surgery.	+ suitable for deep margin assessment (i.e., wound bed inspection) + provides real-time information + can be used to scan large surfaces for occult tumors + Near-infrared (NIR) FLI has high penetration depth (up to 10 mm) − requires administration of fluorescent agents − tumor-specific fluorescent agents have not been studied in laryngeal cancer yet
**Contact** **Endoscopy** **(CE)**	After staining of superficial mucosal cells with methylene blue, a microscopic endoscope is placed in direct contact with the surface, distinguishing tumor from healthy cells in vivo.	+ Suitable to detect residual or recurrent cancer after radiotherapy − not suitable for deep margin assessment − requires topical staining − requires direct tissue contact − steep learning curve for image interpretation
**Optical Coherence Tomography** **(OCT)**	OCT is based on changes in refractive index of tumor cells, by detecting light that is backscattered off tissue boundaries. It is thus capable of imaging cross-sectional anatomy at high resolution in living tissue.	+ provides real-time information + does not require exogenous agents − maximum penetration depth of ~2 mm − requires direct tissue contact
**Raman** **Spectroscopy** **(RS)**	RS uses spectral differences between normal and malignant tissue, and is capable of near-instant, accurate and non-invasive analysis of a tissue’s molecular composition.	+ able to characterize tissues other than mucosa + suitable for deep margin assessment + high specificity + does not require exogenous agents − limited to point measurements − requires direct tissue contact − does not provide real-time information (although near-instant) − Intraoperative RS for margin assessment in laryngeal cancer has not been studied yet

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
