# Peer review of "Current Intraoperative Imaging Techniques to Improve Surgical Resection of Laryngeal Cancer: A Systematic Review"

_cancers, 2021, doi:10.3390/cancers13081895_

Round 1

Reviewer 1 Report

The authors in work entitled:“ Current intraoperative imaging techniques to improve surgical resection of laryngeal cancer: a systematic review“ provide an overview of optical imaging techniques used during laryngeal surgery. The authors performed a systematic search and extraction of data specific to clinical studies, intraoperative techniques, laryngeal cancer surgery and/or diagnostics.

The authors focused on narrow-band, autofluorescence and fluorescence imaging and contact endoscopy. In addition, Raman spectroscopy was mentioned in the discussion as another possibility for detecting cancer tissue in laryngeal cancer.

The review presents current knowledge in the field of detection and surgery for laryngeal cancer. The results and discussion are well written. The advantage and pitfalls of these techniques are discussed.

I have two minor comments:

  1. It would be interesting for reader to state the emission wavelengths also in Figure 1. The emission is now presented by colours, but the wavelength values will be more accurate, because the excitation is already given.
  2. Page 5 last sentence: It is written that “NBI compared to their historical control group”. Could the author better specify the term historical? Is it supposed to be a histological sample?

Reviewer 2 Report

This a systematic review by Lauwerends et al of Erasmus MC Cancer Institute and Isala Hospital in  Netherlands. The study review the current state of utility of intraoperative imaging techniques in patient with laryngeal cancer. Overall, the study design and methodology are sound. The manuscript is well organized and presented. The interpretation and conclusion are confined to the data presented.

Major issues

The main caveat about this study, and as mentioned by the authors in the introduction, is we do not have a strong evidence demonstrating that negative surgical margins are associated with improved outcomes in patients with laryngeal cancer.  

The studies included in the review are generally of small sample size and of diverse clinical presentation, hindering developing clear conclusion about the modalities discussed. The advantage and disadvantage of each modality is somehow fragmented between the results and discussion section. As a suggestion, the authors can synthesis a table summarizing the application of each modality, its advantage, and its disadvantage to convey a clear message about those modality prospect.

In reviewing each of the modality, there is no reporting of validity measures (sensitivity, specificity, PPV, NPV, accuracy) – except for some information in Table 2 - . It is insufficient to report that there was a “significant” or “improvement” in those measures without actually reporting those numbers. For example, Page 7 section 3.2, the authors are reporting “In the clinical studies evaluating the intraoperative benefit of AF for margin as-sessment, sensitivity was consistently reported to be markedly higher than with WL alone [29, 48]” What was that sensitivity? Is it clinically significant?

Minor issues:

Page 5 - Abbreviation TLM mentioned for the first time without previous explanation.

Round 2

Reviewer 2 Report

The authors have addressed my comments sufficiently.